# Predicting Complete Cytoreduction with Preoperative [^18^F]FDG PET/CT in Patients with Ovarian Cancer: A Systematic Review and Meta-Analysis

**DOI:** 10.3390/diagnostics14161740

**Published:** 2024-08-10

**Authors:** Csaba Csikos, Péter Czina, Szabolcs Molnár, Anna Rebeka Kovács, Ildikó Garai, Zoárd Tibor Krasznai

**Affiliations:** 1Division of Nuclear Medicine and Translational Imaging, Department of Medical Imaging, Faculty of Medicine, University of Debrecen, H-4032 Debrecen, Hungary; csikosc@med.unideb.hu (C.C.); czinapeter@gmail.com (P.C.); garai@internal.med.unideb.hu (I.G.); 2Gyula Petrányi Doctoral School of Clinical Immunology and Allergology, Faculty of Medicine, University of Debrecen, H-4032 Debrecen, Hungary; 3Department of Obstetrics and Gynaecology, Faculty of Medicine, University of Debrecen, H-4032 Debrecen, Hungary; molnar.szabolcs@med.unideb.hu; 4Scanomed Ltd., H-4032 Debrecen, Hungary

**Keywords:** ovarian cancer, cytoreductive surgery, preoperative [^18^F]FDG PET/CT, complete debulking, prediction

## Abstract

The cornerstone of ovarian cancer treatment is complete surgical cytoreduction. The gold-standard option in the absence of extra-abdominal metastases and intra-abdominal inoperable circumstances is primary cytoreductive surgery (CRS). However, achieving complete cytoreduction is challenging, and only possible in a selected patient population. Preoperative imaging modalities such as [^18^F]FDG PET/CT could be useful in patient selection for cytoreductive surgery. In our systematic review and meta-analysis, we aimed to evaluate the role of preoperative [^18^F]FDG PET/CT in predicting complete cytoreduction in primary and secondary debulking surgeries. Publications were pooled from two databases (PubMed, Mendeley) with predefined keywords “(ovarian cancer) AND (FDG OR PET) AND (cytoreductive surgery)”. The quality of the included studies was assessed with the Prediction model Risk Of Bias Assessment Tool (PROBAST). During statistical analysis, MetaDiSc 1.4 software and the DerSimonian–Laird method (random effects models) were used. Primary and secondary cytoreductive surgeries were evaluated. Pooled sensitivities, specificities, positive predictive values (PPVs), and negative predictive values (NPVs) were calculated and statistically analyzed. Results were presented in forest plot diagrams and summary receiver operating characteristic (SROC) curves. Overall, eight publications were included in our meta-analysis. Four publications presented results of primary, three presented results of secondary cytoreductions, and two presented data related to both primary and secondary surgery. Pooled sensitivities, specificities, and positive and negative predictive values were the following: in the case of primary surgeries: 0.65 (95% CI 0.60–0.71), 0.73 (95% CI 0.66–0.80), 0.82 (95% CI 0.77–0.87), 0.52 (95% CI 0.46–0.59); and in the case of secondary surgeries: 0.91 (95% CI 0.84–0.95), 0.48 (95% CI 0.30–0.67), 0.88 (95% CI 0.81–0.93), 0.56 (95% CI 0.35–0.75), respectively. The PPVs of [^18^F]FDG PET/CT proved to be higher in cases of secondary debulking surgeries; therefore, it can be a valuable predictor of complete successful secondary cytoreduction.

## 1. Introduction

Ovarian cancer is the second most common type of gynecologic malignancy and the leading cause of gynecological cancer-associated deaths, with 21,750 new cases estimated in 2020 in the United States [1]. Since the disease does not present with early symptoms, and can easily spread in the peritoneal cavity, in more than 70% of the cases it is diagnosed at an advanced stage, where the chance of five-year survival is only approximating 48% [2]. The International Federation of Gynecology and Obstetrics (FIGO) staging system classifies patients with ovarian cancer into four stages based on the size and extent of the tumor [2,3].

The most common histological type of EOC is the high-grade serous type, which represents 75% of all EOC cases [3]. High-grade serous and endometrioid tumors usually present with a high uptake of [^18^F]fluoro-2-deoxy-D-glucose ([^18^F]FDG), while the [^18^F]FDG accumulation of clear cell and mucinous histological subtypes are usually much lower [4].

Cytoreductive surgery (CRS), also known as debulking surgery, is considered to be the backbone of the therapeutic management of EOC. The main goal of the operation is the removal of the entire visible macroscopic tumor, even with the resection of organs if necessary, achieving complete cytoreduction. Preferably, the cytoreductive operation should be performed at the beginning of the therapy (primary cytoreductive surgery), but if inoperability criteria are detected or there is a low chance of achieving complete tumor reduction, the operation is performed after neo-adjuvant chemotherapy (interval debulking surgery). Surgery may also be an option in selected cases after tumor recurrence, in the form of secondary cytoreduction. The main independent prognostic factor in cases of secondary cytoreduction is also complete cytoreduction, which is associated with both better progression-free survival (PFS) and overall survival (OS) [5]. Furthermore, it can still be curative if combined with chemotherapy [6]. The completeness of CRS is usually determined by the evaluation of the operating surgeon at the end of the operation. Complete cytoreduction is declared when no visible tumor can be detected in the abdominal cavity or elsewhere at the end of the operation [7,8]. Overall, complete cytoreduction is the most important prognostic factor in advanced-stage ovarian cancer, but achieving it is a challenging goal, which makes patient selection for these operations essential, including not only imaging and laboratory parameters but also invasive methods such as laparoscopy [5,9].

Current guidelines recommend the use of adjuvant (postoperative) chemotherapy for all patients in stages above Ia. Still, the use of neoadjuvant chemotherapy (NACT) is limited to those patients in whom incomplete debulking has been predicted, or the risk of primary surgery is unacceptably high due to the patient’s general condition [10,11]. Hyperthermic intraperitoneal chemotherapy (HIPEC) is a treatment option when a warm chemotherapeutic solution is circulated in the peritoneal cavity after cytoreductive surgery. The results of this method are promising, yet previous studies have proven that the completeness of cytoreduction strongly limits the success of HIPEC, which underlines the importance of surgery [12].

Several attempts have been made to determine specific predictors—including imaging modalities, tumor markers, and laparoscopic scores—to identify patients eligible for complete cytoreduction [13,14]. Recent guidelines in ovarian cancer management indicate the use of [^18^F]FDG PET/CT for initial diagnosis, staging, prognosis prediction, treatment planning, relapse detection, and therapy assessment. According to these guidelines, the use of PET/CT in treatment planning, including resectability evaluation, is not yet convincing since there are only limited retrospective studies published in this topic [15]. In providing information about affected lymph nodes and peritoneal metastases PET/CT can outperform conventional imaging modalities like CT [15,16]. PET/CT can also provide valuable information about metabolically active metastases and volumetrics like metabolic tumor volume (MTV) or total lesion glycolysis (TLG), which may influence treatment planning and help surgeons achieve complete tumor resection. [^18^F]FDG PET/CT could help in patient selection and tumor resectability evaluation; however, there is still no consensus on the accurate prediction for complete debulking surgery [11]. Limited evidence indicates the need for further studies in this topic.

In our systematic review and meta-analysis, we aimed to evaluate the value of preoperative [^18^F]FDG PET/CT in the prediction of complete cytoreduction in ovarian cancer patients by comprehensively reviewing all existing studies on this topic, and quantitatively assessing the predictive values of preoperative [^18^F]FDG PET/CT for complete cytoreduction.

## 2. Materials and Methods

### 2.1. Eligibility Criteria

The inclusion criteria for our systematic review were the following:enrolled patients were diagnosed with ovarian cancer;patients underwent primary or secondary cytoreductive surgery;patients had preoperative FDG-PET/CT performed prior to cytoreductive surgeries;patients could be categorized into favorable and unfavorable groups according to their PET/CT results, and complete vs. incomplete cytoreductive surgeries could be distinguished;the number of true positive (TP), false positive (FP), true negative (TN), and false negative (FN) cases can be determined from the articles.Grouping the cases into these four categories was performed using the following definitions:
TP = PET/CT favorable and complete cytoreduction achieved;TN = PET/CT unfavorable and incomplete cytoreduction achieved;FP = PET/CT favorable and incomplete cytoreduction achieved;FN = PET/CT unfavorable and complete cytoreduction achieved.

Complete cytoreduction was defined as no macroscopic visible tumor at the end of the surgery.

The exclusion criteria for our systematic review were the following:all articles that were reviews, guidelines, case reports, clinical trials, preclinical studies, and poster abstracts were excluded;articles that used radiopharmaceuticals other than FDG;data from prediction models that included PET/CT results alongside other laboratory and clinical results (e.g., CA-125, HE-4);articles that categorized cases as optimal debulking (less than 1 cm residual tumor diameter) and suboptimal debulking (more than 1 cm residual tumor diameter).

### 2.2. Search Strategy

Two widely used and readily available databases were used to find eligible articles: Mendeley and PubMed. In both databases, a literature search was conducted using the following keywords: “(ovarian cancer) AND (FDG OR PET) AND (cytoreductive surgery)”. The literature search was carried out in September 2023.

All the articles found were screened. After the removal of duplicates, the articles meeting any of our exclusion criteria mentioned above were also removed. After screening the titles and abstracts, those not matching our field of interest were excluded. After reviewing the full texts of the remaining studies, those with sufficient data were included in the end.

Studies were screened and data were collected by two reviewers independently, and discrepancies were resolved by discussing them with a board-certified radiologist and nuclear medicine physician.

### 2.3. Data Extraction

Data on primary and secondary cytoreductive surgeries were analyzed separately. Sensitivities were calculated as “TP/(TP + FN)”, specificities were calculated as “TN/(TN + FP)”, positive predictive values (PPVs) were calculated as “TP/(TP + FP)”, and negative predictive values (NPVs) were given as “TN/(TN + FN)”, using the variables defined above.

### 2.4. Quality Assessment

Quality assessment was completed using the Prediction model Risk of Bias Assessment Tool (PROBAST) [17]. Two separate reviewers assessed each study independently and discrepancies were resolved by consensus.

### 2.5. Statistical Analysis

After data were extracted from the included articles, Meta-DiSc 1.4 software was used for statistical analysis [18]. The DerSimonian–Laird method (Random effects models) was used in data analyses. Data on primary and secondary cytoreduction were analyzed separately. Pooled sensitivities, specificities, and positive and negative predictive values were calculated and visualized in forest plots. Heterogeneity was assessed by calculating Chi-squared and inconsistency index (I^2^). Summary receiver operating characteristic (SROC) curves were formulated utilizing the estimated values of sensitivity, specificity, and their corresponding variances. A threshold effect due to different cut-off values was observed, so Q-indices (the maximum joint sensitivity and specificity) were employed along with area under the curve (AUC) results.

### 2.6. PRISMA Statement

Our systematic review was conducted according to the PRISMA 2020 statement guidelines [19].

## 3. Results

### 3.1. Study Selection and Characteristics

After screening the two databases (Mendeley and PubMed), a total number of 145 results were found. By removing all duplicates, 101 articles were left. These articles were screened based on their titles and abstracts. Publications other than original publications were excluded. Six reviews, four guidelines, eight case reports, three clinical trials, three non-FDG studies, and a poster abstract were excluded. Fifty-three articles were excluded for investigating an unrelated issue. A full-text review of the remaining 23 articles was carried out, and 15 of them were excluded due to insufficient data or for not matching our field of interest. An article was excluded because it assessed optimal instead of complete cytoreduction. Furthermore, this study did not provide enough patient data about their univariate analyses, and the score they created included not only PET/CT parameters but also the Eastern Cooperative Oncology Group performance status (ECOG PS) of the patients [20]. From the studies that assessed a score based on PET/CT and other laboratory parameters together, we included the most significantly associated PET/CT parameter only. We did not include any results influenced by non-[^18^F]FDG PET/CT-related data [21,22]. A total number of eight publications could be included in our systematic review [21,22,23,24,25,26,27,28]. A flowchart summary of the study selection process can be seen in Figure 1.

Characteristics of the included articles can be seen in Table 1. A total number of 620 patients’ data were analyzed from the included studies. Studies were heterogeneous regarding the variables used to categorize PET/CT results as favorable and unfavorable. Primary cytoreductive surgeries were investigated in six of the eight studies. Four studies included data about secondary cytoreduction. Seven studies analyzed patient data retrospectively and one used prospective study methods. Tsoi et al. did not analyze their data of primary and secondary cases separately, only together. Therefore, we conducted our meta-analyses without those data as well. These results are presented in the Appendix A.

Most patients in these studies had epithelial ovarian cancer (96.1%). The majority of these cases belonged to the serous subtype. A detailed summary of the histological subtypes of ovarian cancer cases in the studies included can be seen in Table 2.

The majority of the patients involved had a FIGO stage III (64.2%) or stage IV (16.6%) disease. Details of the studies are found in Table 3.

### 3.2. Quality Assessment

Quality assessment was completed using PROBAST. The risk of bias in each study is presented in Table 4.

### 3.3. Predictive Performance of [^18^F]FDG PET/CT

Data used from the included studies, along with the sensitivities, specificities, and positive and negative predictive values calculated from these data, can be seen in Table 5.

#### 3.3.1. Sensitivity

During the separate analysis of those cases with primary debulking, a pooled sensitivity of 0.65 (95% CI 0.60–0.71) was detected. The pooled sensitivity of the secondary cytoreductive cases turned out to be higher, at 0.91 (95% CI 0.84–0.95). All three syntheses resulted in significant heterogeneity and inconsistency values seen in the forest plots (Figure 2).

The exclusion of the article by Tsoi et al. [27] resulted in slightly different sensitivity values of primary and secondary cytoreduction: 0.61 (95% CI 0.55–0.67) and 0.87 (95% CI 0.78–0.93), respectively (Appendix A).

#### 3.3.2. Specificity

Pooled specificities of primary and secondary cytoreductive cases are presented in Figure 3. In the case of the primary cytoreductive surgeries, PET/CT showed a specificity of 0.73 (95% CI 0.66–0.80) with significant heterogeneity and high inconsistency. In cases of secondary cytoreduction, we calculated a pooled specificity of 0.48 (95% CI 0.30–0.67) with non-significant heterogeneity and very low inconsistency.

With the exclusion of the article by Tsoi et al., a pooled sensitivity of 0.76 (95% CI 0.68–0.82) and of 0.55 (95% CI 0.32–0.77) was calculated for the primary and secondary cytoreductive cases, respectively (Appendix A).

#### 3.3.3. Positive Predictive Value

The pooled PPV of the studies assessing primary debulking surgeries was 0.82 (95% CI 0.77–0.87). For successful secondary cytoreduction, the pooled PPV of PET/CT was higher, with a value of 0.88 (95% CI 0.81–0.93). Even though heterogeneity and inconsistency were high in the analysis of primary surgeries, in the case of the secondary debulking surgeries, a favorable PET/CT could predict complete cytoreduction with non-significant heterogeneity and a low value of inconsistency (Figure 4).

The exclusion of the results by Tsoi et al. [27] changed these results only minimally, giving a pooled PPV of 0.82 (95% CI 0.76–0.87) for primary and 0.90 (95% CI 0.81–0.95) for secondary cytoreduction. Heterogeneity was not significant and inconsistency was low among studies with secondary surgeries (Appendix A).

#### 3.3.4. Negative Predictive Value

Negative predictive values were analyzed similarly, and the results are presented in Figure 5. The pooled NPV of primary cytoreductive cases was 0.52 (95% CI 0.46–0.59) and that of secondary cytoreductive cases was 0.56 (95% CI 0.35–0.75). Heterogeneity was significant in both primary and secondary cases and the inconsistency values were high (Figure 5).

After the exclusion of the results of Tsoi et al. [27], the NPVs were the following: 0.52 (95% CI 0.45–0.58) and 0.48 (95% CI 0.27–0.69) for primary and secondary cytoreductive cases, respectively (Appendix A).

#### 3.3.5. Summary Receiver Operating Characteristic (SROC) Curves

Since false positive cases were relatively rare in our data set, and therefore positive predictive values reached high percentages, a favorable preoperative PET/CT result is likely to predict complete cytoreduction. This is more likely to be true for secondary debulking surgeries since heterogeneity and inconsistency values were low when the different studies were pooled. Heterogeneity between studies and high inconsistency values seen in the rest of our analyses are most likely present because the included studies measured different PET/CT parameters with different cut-off values, as seen in Table 1. Differences in the cut-off values resulting in threshold effects can be followed in the SROC curves (Figure 6). The calculated SROC curves after the exclusion of data from the study of Tsoi et al. are presented in Appendix A. The low negative predictive values experienced are due to the large number of false negative cases. This means that in cases of an unfavorable preoperative PET/CT, complete cytoreduction might still be achievable.

## 4. Discussion

Ovarian cancer is one of the most aggressive types of gynecological malignancies. The cornerstone of therapy is cytoreductive surgery, but complete removal of the entire visible tumor is necessary for the beneficial effects of these operations. This has resulted in the extension of surgical procedures in the past decade, resulting in multi-organ resections if necessary. These operations are demanding for both the patients and the operative facilities throughout the world. Not only specially trained operative staff members and equipment are necessary for these operations, but the availability of operating room time and intensive care unit beds also [29]. For adequate planning, the selection of patients for future successful surgeries is essential; however, predicting complete debulking is a challenging task. [^18^F]FDG PET/CT is one of the few modalities that could potentially be useful in the prediction of future resectability.

In contrast to purely anatomical imaging modalities—like CT or MRI—PET/CT is a diagnostic tool that is not only helpful in visualizing the anatomical relationship between the abdominal organs and tumor implants, but thanks to its high sensitivity to alterations in radiopharmaceutical concentrations it can also detect metabolic changes even before macroscopic changes occur. Metabolically active lymph nodes and peritoneal metastases can therefore be detected earlier in PET than in MRI or CT images [30,31]. This unique feature of PET/CT may further optimize surgical planning.

Our systematic review and meta-analysis aimed to summarize the data available in the literature on how [^18^F]FDG PET/CT is able to predict complete debulking in primary and secondary cytoreduction in ovarian cancer. The role of PET/CT in interval debulking surgeries has not been investigated thoroughly, but it is a promising direction for further investigation. The pooled sensitivities, specificities, and positive and negative predictive values of the eight studies included in our systematic review were calculated. The sensitivity and specificity of primary cytoreductive cases were 65% and 73%, respectively. In the cases of secondary debulking, the sensitivity was higher, while the specificity was lower, being 91% and 48%, respectively. This may be a result of the different study designs and cut-off point selection, as seen from the SROC curves of Figure 6. In relapsed cases of ovarian cancer, the selection of patients for surgery versus chemotherapy is an even more complex problem, and the beneficial effects are even more limited [32]. Positive predictive values were the highest of all values, since the pooled PPVs were 82% and 88% for primary and secondary cytoreductive cases, respectively. Due to the large number of false-negative cases, negative predictive values were lower: 52%, and 56% for primary and secondary cytoreductive cases, respectively.

There is no conventional and generally accepted variable with a cut-off value to say whether a PET/CT result is favorable or not in terms of successful cytoreduction; therefore, every study tries to establish its own prediction model. Significant heterogeneity and inconsistency values were present among the studies included in our meta-analysis, which might be the cause of the different PET/CT parameters with different cut-off values included in the studies assessed. An exception was the analysis of the PPVs of secondary cytoreductive surgeries, where non-significant heterogeneity and a very low inconsistency were seen.

According to our results, a favorable preoperative PET/CT result can predict complete cytoreduction with great probability in those who undergo secondary cytoreduction. Unfortunately, seven of the eight studies included in our meta-analysis analyzed their data retrospectively and only one of the studies used prospective methods. Our study aimed to assess the predictive value of PET/CT alone, and therefore, all the articles that used other modalities such as laboratory parameters to predict complete cytoreduction were excluded. The use of these other laboratory values alongside PET/CT might be able to further improve the predictive values in the future, but these results need further investigation. Also, clinical and laboratory data are essential to follow up with patients and monitor disease recurrence. If clinical parameters suggest the possibility of recurrence, PET/CT might be indicated before surgery to help determine the potential for resectability. Our results suggest that PET/CT could become a useful tool in the prediction of complete secondary cytoreduction in ovarian cancer patients. Even in the modern era of PARP inhibitors, the most important prognostic factor is complete resectability in cases of recurrent ovarian cancer [33]. Nevertheless, further investigation and prospective study models are still needed to develop a standardized image evaluation model that is reliable in discriminating favorable and unfavorable PET/CT results, and thus, can aid patient selection by predicting the completeness of secondary debulking before surgery. Implementation of such a standard could be achieved by utilizing quantitative lesion-based (e.g., number of peritoneal lesions, number of lymph nodes involved, etc.) and metabolic parameters (SUV, MTV, TLG).

### Strengths and Limitations

The strength of our publication is that it is the first meta-analysis investigating the usefulness of preoperative PET/CT imaging in predicting the outcome of primary and secondary cytoreductive surgeries. The limitations of our results include the heterogeneity of the selected studies, especially in those with primary debulking surgeries, and the ratio of lower-stage cases included in some of the studies. For this reason, we also calculated our results with the exclusion of the data from the study by Tsoi et al., but it has only changed our results minimally. Also, a limitation is that all studies but one were retrospective, which emphasizes the need for further prospective studies. We hope that such investigations will be encouraged by our results, since according to the pooled results of the available studies, [^18^F]FDG PET/CT scan is a promising tool for patient selection, especially preceding secondary cytoreductive surgeries.

## 5. Conclusions

Our results showed that the PPV of PET/CT was the highest of the predictive values, especially in the case of secondary debulking surgeries. Our results therefore suggest that the favorable result of a preoperative [^18^F]FDG PET/CT scan is a good predictor of successful secondary cytoreduction in patients with recurrent ovarian cancer. In cases of primary cytoreduction, the PPV is the highest of the predictive values of a PET/CT scan as well, but the available studies show much higher heterogeneity.

## Figures and Tables

**Figure 1 diagnostics-14-01740-f001:**
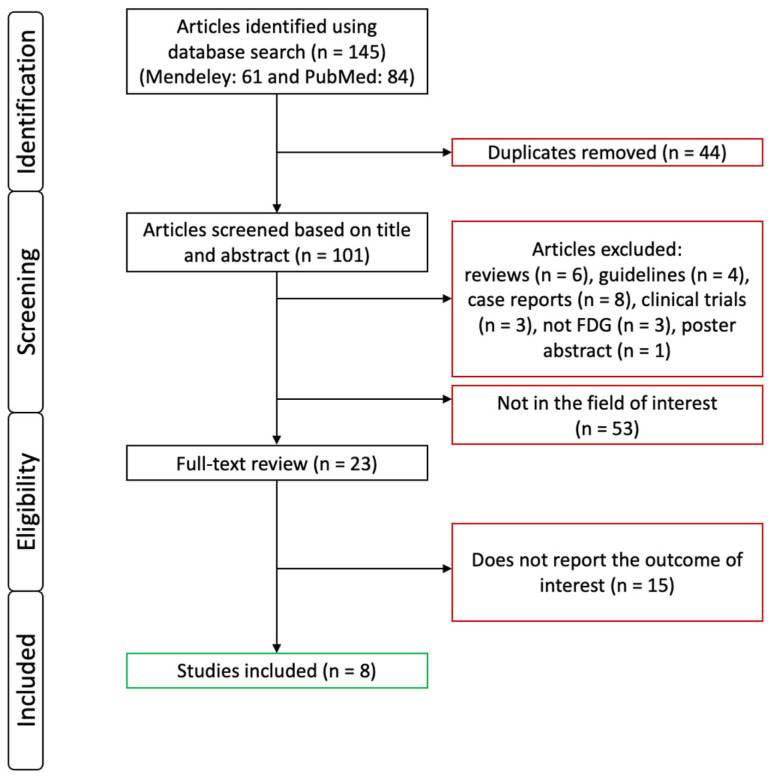
Flow diagram of literature search and study selection summary.

**Figure 2 diagnostics-14-01740-f002:**
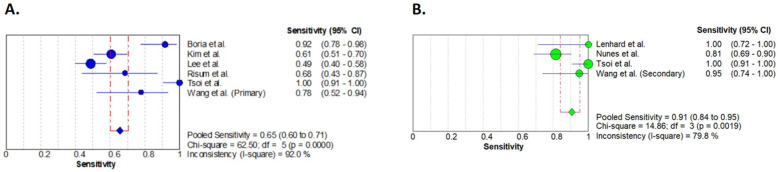
Forest plot analyses of sensitivities. Primary (**A**) and secondary (**B**) cytoreductive cases are presented separately. Results of the individual studies are represented by the circles. Circle sizes represent the weight of the studies. Pooled sensitivities are shown by the squares. Horizontal lines indicate the confidence intervals [21,22,23,24,25,26,27,28].

**Figure 3 diagnostics-14-01740-f003:**
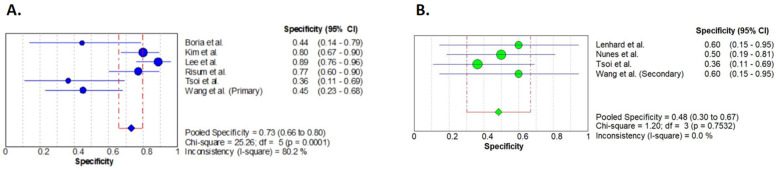
Forest plot analyses of specificities. Primary (**A**) and secondary (**B**) cytoreductive cases are presented separately. Results of the individual studies are represented by the circles. Circle sizes represent the weight of the studies. Pooled specificities are shown by the squares. Horizontal lines indicate the confidence intervals [21,22,23,24,25,26,27,28].

**Figure 4 diagnostics-14-01740-f004:**
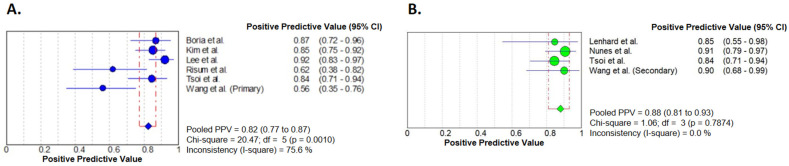
Forest plot analyses of positive predictive values. Primary (**A**) and secondary (**B**) cytoreductive cases are presented separately. Results of the individual studies are represented by the circles. Circle sizes represent the weight of the studies. Pooled PPVs are shown by the squares. Horizontal lines indicate the confidence intervals [21,22,23,24,25,26,27,28].

**Figure 5 diagnostics-14-01740-f005:**
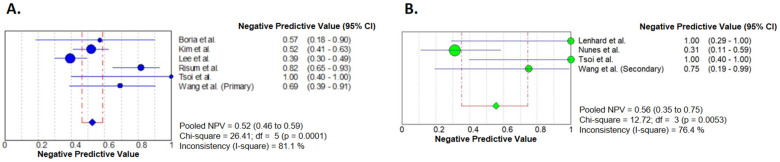
Forest plot analyses of negative predictive values. Primary (**A**) and secondary (**B**) cytoreductive cases are presented separately. Results of the individual studies are represented by the circles. Circle sizes represent the weight of the studies. Pooled NPVs are shown by the squares. Horizontal lines indicate the confidence intervals [21,22,23,24,25,26,27,28].

**Figure 6 diagnostics-14-01740-f006:**
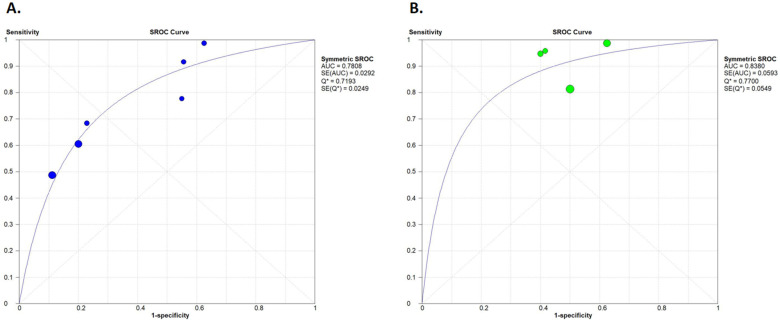
Summary receiver operating characteristic (SROC) curves. Primary cytoreductive cases (**A**) are separated from secondary cytoreductive cases (**B**). Results of the individual studies are represented by the circles. Circle sizes represent the weight of the studies. Q* indicates the point in which the value of sensitivity equals to the value of specificity [21,22,23,24,25,26,27,28].

**Table 1 diagnostics-14-01740-t001:** Characteristics (year published, number of patients, measured PET/CT parameter, type of surgery, study type) of the enrolled articles. In the article published by Wang et al. [28], the number of patients undergoing primary and secondary debulking was reported separately indicated by the arrows.

Study	Year	No. of Patients	Variable Measured	Surgery	Study Type
Boria et al. [21]	2022	45	Extra-abdominal lymph node involvement	Primary	Retrospective
Kim et al. [22]	2023	159	MTV in epigastric and hypochondriac regions	Primary	Retrospective
Lee et al. [23]	2014	166	TLG	Primary	Retrospective
Lenhard et al. [24]	2008	16	PET/CT read	Secondary	Retrospective
Nunes et al. [25]	2023	69	Number of lesions	Secondary	Retrospective
Risum et al. [26]	2008	54	Large bowel mesentery implants	Primary	Prospective
Tsoi et al. [27]	2020	49	Number of FDG-avid peritoneal sites	Primary and Secondary	Retrospective
Wang et al. [28] ➥ Primary ➥ Secondary	2022	62 38 24	MTV	Primary and Secondary	Retrospective

**Table 2 diagnostics-14-01740-t002:** Histological subtypes of ovarian cancer cases in the included studies are presented separately and pooled. The number of cases can be seen in the second column, and their proportional values to the total number of cases in percentages are seen in the third column of each study.

Boria et al. [21]			Kim et al. [22]			Lee et al. [23]				Lenhard et al. [24]		
Epithelial	41	91.1%	Epithelial	149	93.7%	Epithelial		163	98.2%	Epithelial	16	100.0%
High-grade serous	35	77.8%	Serous	118	74.2%	Serous		110	66.3%	All	16	
Low-grade serous	2	4.4%	Endometrioid	12	7.5%	Endometrioid		18	10.8%			
Clear cell	3	6.7%	Mucinous	5	3.1%	Mucinous		16	9.6%			
Endometrioid	1	2.2%	Clear cell	14	8.8%	Clear cell		19	11.4%			
Other	4	8.9%	Other	10	6.3%	Other		3	1.8%			
All	45		Mixed	9	5.7%	Mixed		3	1.8%			
			Other	1	0.6%	All		166				
			All	159								
Nunes et al. [25]			Risum et al. [26]			Tsoi et al. [27]				Wang et al. [28]		
Epithelial	65	94.2%	Epithelial	53	98.1%	Epithelial		43	87.8%	Epithelial	62	100.0%
High-grade serous	54	78.3%	Serous	50	92.6%	Other		6	12.2%	Serous	62	100.0%
Low-grade serous	3	4.3%	Mucinous	2	3.7%	Germ cell		4	8.2%	All	62	
Clear cell	4	5.8%	Endometrioid	1	1.9%	Stromal cell		2	4.1%			
Endometrioid	4	5.8%	Other	1	1.9%	All		49				
Other	4	5.8%	Transitional cell	1	1.9%							
Mixed	3	4.3%	All	54								
Carcinosarcoma All	1 69	1.4%										
				All								
				Epithelial		596	96.1%					
				Other		24	3.9%					
				All		620						

**Table 3 diagnostics-14-01740-t003:** FIGO stages of the patients enrolled in the included articles separately and pooled. The number of cases can be seen in the first column, and their proportional values to the total number of cases in percentages are seen in the second column of each study.

Stage	Boria et al. [21]		Kim et al. [22]		Lee et al. [23]		Lenhard et al. [24]		Nunes et al. [25]		Risum et al. [26]		Tsoi et al. [27]		Wang et al. [28]		All	
FIGO I.					65	39.2%			7	10.1%			15	30.6%			100	16.1%
FIGO II.									1	1.4%			12	24.5%				
FIGO III.	36	80.0%	115	72.3%	87	52.4%			55	79.7%	50	92.6%	18	36.7%	37	59.7%	398	64.2%
FIGO IV.	9	20.0%	44	27.7%	14	8.4%			6	8.7%	4	7.4%	1	2.0%	25	40.3%	103	16.6%
Unknown							16	100.0%					3	6.1%			19	3.1%
All	45		159		166		16		69		54		49		62		620	

**Table 4 diagnostics-14-01740-t004:** Risk of bias and applicability assessment of the included articles. Green dots indicate a low risk of bias and low concerns of applicability, yellow dots indicate an unclear risk of bias, and red dots indicate a high risk of bias and high concerns of applicability.

Study	Risk of Bias					Applicability			
	Participants	Predictors	Outcome	Analysis	Overall	Participants	Predictors	Outcome	Overall
Boria et al. [21]	Unclear ◉	Low ◉	Low ◉	Low ◉	Unclear ◉	Low concerns ◉	Low concerns ◉	Low concerns ◉	Low concerns ◉
Kim et al. [22]	Low ◉	Low ◉	Low ◉	Low ◉	Low ◉	Low concerns ◉	Low concerns ◉	Low concerns ◉	Low concerns ◉
Lee et al. [23]	Unclear ◉	Low ◉	Low ◉	Unclear ◉	Unclear ◉	Low concerns ◉	Low concerns ◉	Low concerns ◉	Low concerns ◉
Lenhard et al. [24]	Unclear ◉	Unclear ◉	High ◉	High ◉	High ◉	Low concerns ◉	Low concerns ◉	Low concerns ◉	Low concerns ◉
Nunes et al. [25]	Unclear ◉	Low ◉	Low ◉	Low ◉	Low ◉	Low concerns ◉	Low concerns ◉	Low concerns ◉	Low concerns ◉
Risum et al. [26]	Low ◉	Low ◉	Low ◉	Low ◉	Low ◉	Low concerns ◉	Low concerns ◉	Low concerns ◉	Low concerns ◉
Tsoi et al. [27]	High ◉	Low ◉	Low ◉	Low ◉	High ◉	High concerns ◉	Low concerns ◉	Low concerns ◉	High concerns ◉
Wang et al. [28]	Low ◉	Low ◉	Low ◉	Low ◉	Low ◉	Low concerns ◉	Low concerns ◉	Low concerns ◉	Low concerns ◉

**Table 5 diagnostics-14-01740-t005:** Summary of the number of true positive (TP), true negative (TN), false positive (FP), and false negative (FN) cases and the calculated sensitivity, specificity, PPVs, and NPVs of the included studies. In the article published by Wang et al. [28], data were reported separately indicated by the arrows.

Study	TP	FP	TN	FN	Sensitivity	Specificity	PPV	NPV
Boria et al. [21]	33	5	4	3	91.7%	44.4%	86.8%	57.1%
Kim et al. [22]	63	11	44	41	60.6%	80.0%	85.1%	51.8%
Lee et al. [23]	59	5	40	62	48.8%	88.9%	92.2%	39.2%
Lenhard et al. [24]	11	2	3	0	100.0%	60.0%	84.6%	100.0%
Nunes et al. [25]	48	5	5	11	81.4%	50.0%	90.6%	31.3%
Risum et al. [26]	13	8	27	6	68.4%	77.1%	61.9%	81.8%
Tsoi et al. [27]	38	7	4	0	100.0%	36.4%	84.4%	100.0%
Wang et al. [28]	32	13	12	5	86.5%	48.0%	71.1%	70.6%
➥Primary	14	11	9	4	77.8%	45.0%	56.0%	69.2%
➥Secondary	18	2	3	1	94.7%	60.0%	90.0%	75.0%

## Data Availability

The datasets used and analyzed in the current study are available from the corresponding author on reasonable request.

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
