# Peer review of "Predicting Complete Cytoreduction with Preoperative [18F]FDG PET/CT in Patients with Ovarian Cancer: A Systematic Review and Meta-Analysis"

_diagnostics, 2024, doi:10.3390/diagnostics14161740_

Round 1

Reviewer 1 Report (Previous Reviewer 2)

Comments and Suggestions for Authors

The enclosed manuscript is a re-submitted work from Csikos et al. regarding the use of F-18-FDG PET images to provide complete cytoreduction prediction. Eight out of the 145 articles searched on the medical publication search engines were selected for the systemic review. Given the strong rationale, the systemic review analysis provided a reasonable explanation of how and why the SUV, as well as other PET imaging features, can be used in the prediction of ovarian cancer prognosis. The current format is well-established and clearly stated, but a few concerns are provided for further polishing the article for reading-friendly purposes. 

1) Table 2 provides the breakdown of ovarian subtypes among all 8 reports; however, the discrepancies between articles made the table unorganized. The authors may consider using other approaches to address the disagreements between articles. 

2) It would be clearer (in the discussion part) how this approach can be helpful to clinical applications, such as the timing of using PET imaging and the challenges when looking back to some datasets (in the 8 included articles) that show less relevance to the prognostic values. 

Comments on the Quality of English Language

The English is mostly acceptable for reading. 

Author Response

Dear Reviewer,

Thank you very much for taking the time to review this manuscript.  We found your comments very useful and have made corrections in our revised manuscript according to them. Please find the detailed responses below and the corresponding revisions/corrections highlighted/in track changes in the re-submitted files.

Comments 1: Table 2 provides the breakdown of ovarian subtypes among all 8 reports; however, the discrepancies between articles made the table unorganized. The authors may consider using other approaches to address the disagreements between articles.

Response 1: Thank you for pointing this out. For some of the 8 articles used different approaches to categorize the subtypes, we agree that table 2 needed revision, to make it more clear. We have reorganized table 2, and highlighted the 2 major histological categories as “epithelial and others” in case of all 8 articles, and further subdivided these categories to make table 2 more organized. Subcategories remained shown as well for the purpose of completeness. We hope these changes have successfully improved the appearance of table 2., making it easier to comprehend for the reader. (Lines 209-210).

Comments 2: It would be clearer (in the discussion part) how this approach can be helpful to clinical applications, such as the timing of using PET imaging and the challenges when looking back to some datasets (in the 8 included articles) that show less relevance to the prognostic values.

Response 2: Thank you very much for the important suggestion. We implemented extra detail in the discussion part on how this approach of using PET/CT can help clinical decision making and challenges we are facing with. We added to the text that clinical and laboratory data are the most important for patient follow-up and raising the suspicion for disease recurrence. When such event occurs, a preoperative PET/CT can be a helpful modality to provide the extent of the disease and guide the clinicians weather surgery is a feasible option. We also mentioned that PET/CT is often more sensitive and is able to detect metabolic lesions before anatomical imaging modalities (MRI, CT). As far as challenges are concerned, we need further, prospective studies to develop a standardized model for evaluating quantitative PET/CT parameters that could maximize the potential of this modality to predict surgical outcome. Added information in the text regarding the changes mentioned above can be found in lines 303-309, lines 342-354.

Response to Comments on the Quality of English Language

Point 1: The English is mostly acceptable for reading

Response 1: Thank you for your feedback on the quality of English language, we have revised the English of the manuscript again.

Reviewer 2 Report (New Reviewer)

Comments and Suggestions for Authors

Dear authors,

Thanks for your exceptional work

Only a few remarks to be noted

1- Please explain why you chose only two databases instead of three. Also, why was the Mendeley database specifically chosen?

2- Please add a paragraph in your discussion about the added value of PET CT over CT & MRI

3- Please explain in your discussion how the PET CT can practically provide data about the completeness of cytoreduction. This is actually the core of your manuscript which is still not clearly explained. What are the findings the surgeon can use to build his decision?

4- Please explain this sentence in the results: "53 articles not matching our field of interest were excluded as well."

5- Please move the limitation section to the end of the discussion before -not after- the conclusion

Author Response

Dear Reviewer,

Thank you very much for taking the time to review this manuscript, we found your comments very useful and revised our manuscript according to your suggestions. Please find the detailed responses below and the corresponding revisions/corrections highlighted/in track changes in the re-submitted files.

Comments 1: Please explain why you chose only two databases instead of three. Also, why was the Mendeley database specifically chosen?

Response 1: Thank you for your question. We chose the two databases that we thought to be widely used and to contain a large number of medically related articles. Furthermore, these databases are readily available and user friendly as well. Therefore we added information to clarify this in the methods section, and Line 132 was supplemented with this information.

Comments 2: Please add a paragraph in your discussion about the added value of PET CT over CT & MRI

Response 2: Thank you very much to point out the importance of the missing information that is how PET/CT is superior to CT or MRI imaging. We found it very useful and we fully agree with your comment. We added a paragraph to the discussion detailing that PET imaging is a highly sensitive modality that can detect metabolically active lesions in the lymph nodes and on the peritoneal surface before anatomical imaging modalities, such as CT and MRI could be able to detect these (Lines 303-309).

Comments 3: Please explain in your discussion how the PET CT can practically provide data about the completeness of cytoreduction. This is actually the core of your manuscript which is still not clearly explained. What are the findings the surgeon can use to build his decision?

Response 3: Thank you for your comment. We implemented extra detail in the discussion part on how this approach of using PET/CT can help clinical decision making. We added to the text that clinical and laboratory data are the most important for patient follow-up and raising the suspicion for disease recurrence. When such event occurs, a preoperative PET/CT can be a helpful modality to provide the extent of the disease and guide the clinicians whether surgery is a feasible option. However, we need further, prospective studies to develop a standardized model for evaluating quantitative PET/CT parameters that could maximize the potential of this modality to predict surgical outcome. Added information in the text regarding the changes mentioned above can be found in lines 342-354. As far as assessing whether the surgery was complete or not, we have to rely on the operative reports (the operating surgeon’s description) as indicated in lines 67-70 of the introduction. Image guided surgical intervention might improve the assessment of completeness in the future, unfortunately there is not enough data to investigate this in our current study. In other words, for now PET/CT is a potential tool for predicting complete secondary cytoreduction, but not for determining whether completeness has been achieved during surgery. For the evaluation of the completeness of surgeries the operative reports suggested by international guidelines (e.g. ESGO ovarian cancer operative report) should be used. If tumor markers were elevated preoperatively the drop in the serum concentration of these markers could also be an indicator of the completeness of the operation. 

Comments 4: Please explain this sentence in the results: "53 articles not matching our field of interest were excluded as well.

Response 4: Thank you for pointing out that this sentence is not well-explained, we fully agree with the comment. Articles identified by the keywords indicated in the text were screened by reading the titles and abstracts. What we meant by “not matching our field of interest” is that after reading the titles and abstracts these articles were found not to investigate how PET/CT can predict complete cytoreduction. The requested sentence was completed with the explanation in the text as well. We replaced the term "not matching our field of interest" with "excluded for investigating an unrelated issue" (Lines 175-176).

Comments 5: Please move the limitation section to the end of the discussion before -not after- the conclusion

Response 5: Thank you for your suggestion. We agree with your comment that this part need to be part of the discussion session rather than in the conclusion. Therefore, we moved the requested paragraph at the end of the discussion. (Lines 355-366)

This manuscript is a resubmission of an earlier submission. The following is a list of the peer review reports and author responses from that submission.

Round 1

Reviewer 1 Report

Comments and Suggestions for Authors

Thank you for the opportunity to review the revised manuscript.

The authors aim to determine whether complete tumor resection can be achieved using preoperative PET/CT in ovarian cancer patients. They are combining studies on primary debulking surgery and secondary debulking surgery. However, these two surgeries have entirely different objectives and differ in terms of chemotherapy administration. Am I the only one who feels that combining these for evaluation lacks clinical significance? For instance, in the case of secondary debulking surgery, it is known that various factors are related to diagnosis, as shown in the following paper: https://doi.org/10.1007/s00259-013-2610-9.

Comments on the Quality of English Language

None

Reviewer 2 Report

Comments and Suggestions for Authors

The enclosed study presents a comprehensive meta-analysis that underscores the importance of [18F]FDG PET/CT imaging in the preoperative assessment for ovarian cancer cytoreductive surgery. The study is methodologically sound, utilizing a systematic approach to literature review and employing statistical tools to be systemically analyzed. The findings indicate that [18F]FDG PET/CT has a significant role in predicting the success of complete tumor removal in both primary and secondary surgeries, with extreme predictive values for secondary debulking procedures. The high positive predictive values (PPVs) suggest that when preoperative [18F]FDG PET/CT results are favorable, the chances of achieving complete cytoreduction are increased, which can be crucial for patient prognosis and treatment planning. Overall, the article provides valuable insights that could potentially improve clinical decision-making and patient outcomes in ovarian cancer treatment. Despite the draft being well organized, a few minor concerns can be considered for further polishing the article. 

1) Considering the relationship between the FDG uptakes and the prognostic value of cytoreduction surgery, a detailed introduction or discussion might be necessary to highlight the rationale/purposes/findings of this draft. 

2) Something missing is the connections between results and the conclusions. It is apparent that the results are not consistent regarding the small sample size (n=8) and the lack of consistency between studies. All of the revealed data: specificity, sensitivity, PPV, and NPV were all found heterogenous. In such circumstance, it is unclear why the conclusion can be made. More descriptions or clarifications might be needed. 

Comments on the Quality of English Language

Rephrases of the description might be needed to make the article easy to read.